# Quantification of Lumbar Spinal Canal Stenosis by Quantitative Fat-Suppressed Contrast-Enhanced Magnetic Resonance Imaging

**DOI:** 10.3390/jcm9103084

**Published:** 2020-09-24

**Authors:** Yong Jun Jin

**Affiliations:** Department of Neurosurgery, Inje University College of Medicine, Seoul Paik Hospital, Seoul 04551, Korea; jyj0819@hanmail.net; Tel.: +82-10-6251-3729; Fax: +82-2-2270-0467 (ext. +82-2-2270-0030)

**Keywords:** lumbar central stenosis, internal vertebral venous plexus, fat-suppressed T1-weighted enhanced, magnetic resonance images, redundant nerve root, nerve root sedimentation sign

## Abstract

Fat-suppressed T1-weighted magnetic resonance images (MRIs) enhanced with gadolinium can evaluate the internal vertebral venous plexus and cauda equina. This study compared such findings with clinical situations and discusses whether these are helpful for symptomatic grading and selection at the surgical level in patients with lumbar central stenosis. A total of 263 patients (337 levels < 75 mm^2^ of dural cross sectional area (DCSA)) were included. The enhancement patterns of dorsal epidural vein (DVCE), periradicular vein (PVCE) and intraradicular vein (IRCE) were assessed qualitatively. The quantification of IRCE was acquired by the ratio (%) (enhancement parameters: MS/P1, MS/P2, WR/P1, WR/P2) of signal intensities between the cauda equina (MS-IRCE: maximal spot rootlet, WR-IRCE: whole rootlets) and psoas muscle (P1, P2). Receiver-operator characteristic curves were plotted to obtain imaginary cutoff values for the prediction of symptomatic appearance or operation decision. All levels were classified into seven groups on the basis of pain distribution and the presence of IRCE. PVCE was significantly related to high incidences of symptoms, unilaterality and operation. DVCE and IRCE were connected with high incidences of symptoms, bilaterality and operation. IRCE was also related to high visual analogue scale (VAS), small DCSA and high enhancement parameters. The order of the group was concordant with the degree of enhancement parameters (*p* = 0.000). Cutoff values of enhancement parameters for prediction were as follows: symptoms (147/123/140/121), bilaterality (165/139/157/137) and operation (164/139/159/138). Enhancement patterns and parameters could help in stratification, grading and decision-making at the surgical level.

## 1. Introduction

Neurogenic intermittent claudication (NIC) in lumbar stenosis can be explained by the venous congestion theory. Kobayashi et al. explained that the pathogenesis of NIC was composed of intrathecal consecutive events such as mechanical circumferential compression, the occlusion of subarachnoid space, venous congestion, nerve injury, intraradicular edema, conduction disturbance and ectopic discharge [1].

However, extrathecal events should also be considered. The internal vertebral venous plexus (IVVP) in the lumbar spine is located between the spinal bony canal and dura mater (Figure 1). It communicates with the basivertebral vein and external vertebral venous plexus (EVVP) through the intervertebral foramen [2,3]. The intraradicular microvessels drain into the radicular veins, which are connected through the dura with tortuous periradicular extradural venous convolutions (PEVC) and the IVVP. The transdural part of the radicular vein shows extensive venous narrowing at the point of penetration of the dura mater. The radicular veins have intravenous dural folds, undergo a very tortuous and oblique path through the dura where the lumen narrows and contain smooth muscle fibers in the vascular walls in the intradural and extradural sections that could regulate reflux in cases of venous hypertension [4]. Nevertheless, this system has limitations in the presence of spinal vascular pathologies such as arteriovenous fistulas or malformations. This notion is supported by the observation that exertion could make symptoms worse, secondary to uncompensated pressure changes that interfere with circulation [5]. Elevated epidural mechanical pressure at the stenotic level could block not only subarachnoid space, but also the venous flow in PEVC and IVVP. Subsequent venous hypertension could play a role in venous reflux to the radicular vein and intraradicular microvessels. Transdural venous reflux of the radicular vein could lead to intraradicular congestion and edema. The author assumes that the congestion of IVVP could play a crucial role in venous reflux to the intraradicular vessels. The congestion of IVVP and intraradicular vessels should be assessed both qualitatively and quantitatively.

Fat-suppressed T1-weighted magnetic resonance images (MRIs) enhanced with gadolinium (Gd) could show the dilation of microvessels, which is accompanied by inflammatory reactions. This technique could be useful to delineate the venous structure in cases of venous hypertension. Gd could leak through the endothelial fenestration of dilated vessels. The enhanced tissue in the lumbar spinal canal is primarily a venous structure comprising IVVP. The venous plexus can be divided into anterior (the retrocorporeal vein, periradicular or longitudinal vein and intervertebral vein) and posterior (the dorsal epidural or longitudinal vein) IVVP [2,3] (Figure 1). The enhancement of the periradicular or longitudinal vein of the anterior IVVP (PVCE, Figure 2), the dorsal epidural or longitudinal vein of the posterior IVVP (DVCE, Figure 3) and the intraradicular vein of the cauda equina (IRCE, Figure 4) were frequently observed at the site of stenosis. Pain intensity induced by venous congestion could be presumed to increase proportionally depending on the degree of enhancement. Consequently, this technique could be regarded as a functional image representing sites of pain. This study aimed to compare such patterns in clinical situations and to discuss whether these are helpful in symptomatic grading and selection at the surgical level in lumbar spinal stenosis (LSS) patients.

## 2. Materials and Methods

In a prospective, observational study conducted between *March* 2011 and January 2020, 1815 consecutive patients with pain in the midline or lateralized back, buttock and leg underwent Gd-enhanced T1-weighted MRI with fat suppression, including the coronal, sagittal and axial planes in addition to conventional T1, T2-weighted imaging. The present study is under way to examine the relationships between pain distribution and the enhancement of outer annulus, periannular IVVP and facet capsule. All MR images were obtained with the same platform and equipment (a 1.5 T Achieva scanner, Philips Medical Systems, Bests, The Netherlands) in the same hospital to avoid scanner variations among different vendors and to assess the qualitative measurement of enhancement by using a 16-channel phased-array sensitivity-encoding (SENSE) body coil. Turbo-spine echo (TSE), an applied pulse sequence, was selected on the conventional image without enhancement. Patients were placed in a supine position. To localize most stenotic lesions, images on axial planes were obtained in parallel with each disc space. In contrast-enhanced MRI, a SPIR (spectral presaturation with inversion recovery) sequence was used. Images were obtained in a regular sequence on the axial (repetition time (TR): 400 ms, echo time (TE): 7.4 ms, matrix scan: 256, slice thickness: 4 mm, slice gap: 0.4 mm, field of view (FOV): 220 mm, flip angle (FA): 90°, total scan duration: 3 min 8.8 s), coronal (TR: 400 ms, TE: 7.4 ms, matrix scan: 304, slice thickness: 4 mm, slice gap: 0.4 mm, FOV: 320 mm, FA: 90°, total scan duration: 2 min 20 s) and sagittal (TR: 400 ms, TE: 7.4 ms, matrix scan: 288, slice thickness: 4 mm, slice gap: 0.4 mm, FOV: 320 mm, FA: 90°, total scan duration: 2 min 12.8 s) planes. For the elimination of effect on the time-dependent changes showing contrast enhancement in venous structures within a spinal canal, T1-weighted images were acquired 30 s after intravenous administration of gadolinium-based contrast agents at a dose of 0.1 mmol/kilogram of body weight (gadopentetate dimeglumine, Magnevist^®^, Schering, Berlin, Germany (2011–2014); gadoterate meglumine, Uniray^®^, Dongkook Lifescience, Seoul, Korea (2015–2020))

Cases with lumbar stenosis were selected among the original 1815 patients. Lumbar stenosis is composed of central (absolute criteria <75 mm^2^ of the dural cross sectional area (DCSA), relative criteria <100 mm^2^) and lateral stenosis, which is further divided into lateral recess stenosis and foraminal stenosis [1]. Inclusion criteria are central or lateral recess stenosis with DCSA <75 mm^2^, which can initiate a neurological dysfunction. Among cases with DCSA <75 mm^2^, 69 foraminal stenoses, 50 extraforaminal stenoses, five synovial cysts, four epidural lipomatosis and six indeterminate diagnoses were excluded. Cases (*n* = 32) with ipsilateral yellow ligament hypertrophy with DCSA >75 mm^2^ were excluded to rule out the confounding effect of local mechanical and inflammatory responses. Cases (*n* = 215) with bilateral back pain, confirmed by the medial branch block or facet capsule injection, were also excluded irrespective of DCSA. Finally, 263 patients (337 stenotic levels, mean age 67.2 years, range 41–87 years, female 59%) were included in this study. For a control group with age and sex matching, 15 asymptomatic patients (mean age 68.1 ± 5.5 years, range 61–77 years, female 53%) without any stenosis (DCSA = 133.5 ± 40.7 mm^2^ for 30 L3–4 + L4–5 levels) were recruited. This retrospective analysis of a prospective observational study was approved by the institutional review board at the hospital (File number: 2020-05-001) and informed consent was obtained from all patients.

### 2.1. Assessment of DCSA, PVCE, DVCE and IRCE (Whole Rootlets (WR), Maximal Spot Rootlet (MS))

DCSAs were measured on the T2-weighted axial plane at the most stenotic level in all stenosis groups and at the L3–4 and L4–5 levels in the control group. The presence of enhancement patterns such as PVCE, DVCE, and IRCE was assessed qualitatively at the level of stenosis. PVCE indicated the enhancement of the periradicular vein and was regarded as a part of the longitudinal vein in anterior IVVP or PEVC (Figure 2). DVCE designated the enhancement of the dorsal epidural venous plexus in the posterior IVVP (Figure 3). IRCE indicated the enhancement of the intraradicular vein within intrathecal rootlets (Figure 4). The author assumed that these enhancement patterns (PVCE, DVCE, IRCE) developed according to the degree of venous congestion and reflux.

The quantification of IRCE was evaluated by the ratio (%) of signal intensities between the cauda equina and psoas muscle on the same axial enhanced image. The signal intensity of rootlets was measured within the manually defined region of interests (ROIs) along the boundary of the white signal by a digitizer, Maroview software (Infinitt Healthcare Co., Seoul, Korea). To define an exact margin, intrathecal rootlets were marked on the same T2-weighted axial image. The adjustment of the gray scale (window level) was necessary for the discrimination between the rootlets and cerebrospinal fluid. Manual ROI analysis was performed by superimposing previously acquired ROI on the enhanced fat-suppressed T1-weighted image (Figure 5). Two methods were composed of whole rootlets IRCE (WR-IRCE) and maximal spot rootlet IRCE (MS-IRCE) at the most stenotic level (Figure 6). WR-IRCE was obtained by recording the mean signal intensities of all outlined intrathecal rootlets. MS-IRCE was defined as the maximum value in circular ROIs (0.18 mm^2^) among intrathecal rootlets. In some patients, the large radiculomedullary veins showed strikingly higher signal intensities (>315) than those of other rootlets. These high signals were excluded in the subsequent measurements. The measurement of WR-IRCE and MS-IRCE on the L3–4 and L4–5 levels, respectively, were also performed in the asymptomatic control group to find a difference between such levels. The signal intensity measurement of the psoas muscle was performed within manually defined ROIs (Psoas1), which were delineated along the margin of the unilateral psoas muscle on the same enhanced axial images. In addition, circular ROIs (33.42 mm^2^, Psoas2) were designated within the medial portion of the psoas muscle corresponding to a mid to posterior disc space to avoid field inhomogeneity (Figure 7). Using these signal intensities, four rootlet-to-psoas ratios were defined (MS/P1 = MS-IRCE/Psoas1*100, WR/P1 = WR-IRCE/Psoas1*100, MS/P2 = MS-IRCE/Psoas2*100, WR/P2 = WR-IRCE/Psoas2*100). Those parameters were analyzed to identify the best possible candidate to stratify patients with lumbar central stenosis.

### 2.2. Criteria of Grouping

All patients were classified as groups with bilateral buttock or leg pain, unilateral buttock or leg pain, midsacral or lumbosacral junction pain and no symptoms. The author assumed that pain distribution could have a connection with the neuroanatomical location where a venous congestion exists. The bilateral buttock and leg symptom—including neurogenic intermittent claudication—favors the severe venous congestion or edema of intrathecal rootlets, which could lead to the development of a higher IRCE value. Bilateral leg pain could be related to harsher conditions than unilateral leg pain. Pain in the midsacral or lumbosacral junction area could have originated from a less congestive status (low IRCE value) than pain in the leg. Cases with a unilateral leg pain seem to have a localized periradicular venous congestion along the proximal traversing root (the proximal process of dorsal root ganglion) rather than the congestion of intrathecal rootlets. This pathomechanism could be concordant with a typical lateral recess stenosis. Lower IRCE values could be expected in patients with unilateral leg symptoms. However, it is rarely possible that a patient with high IRCE value, which means the congestion of other intrathecal rootlets, has unilateral, but no bilateral leg pain. In terms of the location of venous congestion, it indicated that the unilateral proximal process of dorsal root ganglion was compressed mechanically or congested enough to develop a pain, but other intrathecal rootles were insufficiently compressed mechanically or congestive. This situation could be defined as an advanced state compared to unilateral leg pain with low IRCE. In contrast to symptomatic patients, it was highly anticipated that asymptomatic patients with central stenosis would have less congestion and lower IRCE value. In addition, patients of control group were expected to have the least IRCE values compared with asymptomatic patients with central stenosis.

The presence of visible IRCE is presumed to be necessary in the grouping criteria for an in-depth segmentation of symptomatic patients based on enhanced T1-weighted images with fat suppression. Therefore, the enrolled patients were graded according to the combination of pain distribution and the presence of IRCE. All levels were classified into seven groups ((Group 1): IRCE(+)/bilateral, (Group 2): IRCE(+)/unilateral, (Group 3): IRCE(−)/bilateral, (Group 4): IRCE(−)/midsacral, (Group 5): IRCE(−)/unilateral, (Group 6): IRCE(−)/no Sx, (Group 7): control). The group numbering indicates the order of predictable severity in cases with intrathecal intraradicular venous congestion or edema. In particular, only patients with an effect after an interlaminar epidural steroid injection and without an effect after medial branch block were included in Group 4.

### 2.3. Comparison with Other Radiologic Findings Related to LSS

The presence of degenerative spondylolisthesis (SPL), redundant nerve root (RNR) and nerve root sedimentation sign (NRSS) were checked to compare these with the enhancement ratio to predict the presence or severity of symptoms. Spondylolisthesis could have a relation to a severe enhancement pattern, which is likely to be related to dynamic compression. RNR refers to thickened, tortuous and elongated roots of the cauda equina above the level of a long-standing, severe extradural compression [6]. The deformation of intrathecal rootlets indicates mechanical compression and tethering. Positive RNR could be related to high IRCE. As per the protocol of Barz et al. on NRSS [7], a positive sign was defined as the absence of nerve root sedimentation at the level above or below the level of maximum stenosis. Such radiologic findings were evaluated qualitatively.

### 2.4. Treatment Modalities and the Evaluation of Outcome

Among patients without improvement after medication such as nonsteroidal anti-inflammatory drugs (NSAIDs) and gabapentin, epidural steroid injections were performed to evaluate clinical responses. Interlaminar approaches in patients with DVCE or IRCE and transforaminal approaches (conventional and posterolateral route) in patients with PVCE were preferred according to the enhancement sites and concordant with symptomatic nature. Symptomatic patients received follow-up care for at least two months after injections. Clinical responses were assessed as poor (0–24%), fair (25–49%), good (50–74%) and excellent (75–100%) based on the reduced difference between the visual analog scale (VAS) and the preprocedural VAS, measured at two weeks (visit 1) or one month (visit 2) after injection. Because pain interventions show favorable outcomes only in short-term intervals, all follow-ups were performed within one month. The decision to operate was entirely dependent on the patients’ willingness after two months. The operation rate at the final follow-up was assessed to determine the severity of symptoms among groups. A favorable outcome was defined as good or excellent responses.

### 2.5. Statistical Analysis

Continuous variables, including age, DCSA, VAS back, VAS leg, MS-IRCE, WR-IRCE, Psoas1, Psoas2, MS/P1, WR/P1, MS/P2 and WR/P2, were evaluated. A single rater (YJJ) with more than 17 years of expertise in the spine assessed all MR images twice. An interval of at least one month was used between repeated measurements (the day of MRI and follow-up after one month). Categorical variables were sex, stenotic level, presence of symptoms, laterality of symptoms (bilateral/unilateral/midsacral), patterns of enhancement (PVCE/DVCE/IRCE), treatment modalities (operation/block/medication), presence of SPL/RNR/NRSS and outcomes (excellent/good/fair/poor).

IBM SPSS Statistics for Windows version 24 (IBM Corp, Armonk, NY, USA) was used for all statistical analyses. The average pairwise Cohen’s kappa test was used to determine intraobserver reliability. The Landis and Koch guidelines were used to categorize the kappa value and define the strength of agreement, with values of 0.81–1.00, 0.61–0.80, 0.41–0.60, 0.21–0.40 and 0–0.20 indicating excellent, substantial, moderate, fair and slight agreements, respectively [8].

Continuous variables were examined with *t*-tests, one-way analysis of variance (ANOVA) including Games-Howell multiple comparison test (post hoc analysis), bivariate correlation analysis and linear regression analysis to investigate the possibility of grading or stratification among groups. Additionally, statistical differences between groups were assessed. Correlations between DCSA/age/VAS and enhancement parameters were studied.

Categorical variables were assessed by χ^2^ tests for the confirmation of independency, sensitivity, specificity, positive predictive value (PPV), negative predictive value (NPV), accuracy, odds ratio (OR) and relative risk (RR). Incidences of these variables were evaluated and compared between groups. The correlation of PVCE/DVCE/IRCE with the presence of symptom, laterality of symptom and operation rate was investigated. The incidence of SPL was evaluated to find any relationship with visible IRCE, presence of symptom and the presence of RNR/NRSS. The influence of RNR/NRSS on DCSA, IRCE, VAS, and the presence of symptoms were also studied.

### 2.6. Determination of the Cutoff Values

Receiver-operator characteristic (ROC) curves were plotted to produce imaginary cutoff values with high sensitivity and specificity to determine the superiority of enhancement parameters in comparison with DCSA for the prediction of symptom, bilaterality of symptom, visible IRCE and operation. The most predictable parameter in terms of the area under the curve (AUC) was selected. Cutoff values were determined with appropriate sensitivity and specificity.

## 3. Results

### 3.1. Reliabilities of Variables

The intraobserver reliability of DCSA, MS-IRCE, WR-IRCE, Psoas1, Psoas2, MS/P1, WR/P1, MS/P2, and WR/P2 ranged between 0.82 and 0.95 kappa, while that of DVCE, IRCE, PVCE, DS, RNR, and NRSS ranged between 0.87 and 0.97 kappa. The intraobserver agreement was excellent for all variables.

### 3.2. Stratification and Grading of Patients with Central Stenosis

Intergroup difference in age was only significant in Group 5 (Table 1). The VAS change between groups was not statistically different (*p* = 0.050). DCSA became significantly smaller as the group number decreased (*p* = 0.000). In multiple comparison tests, intergroup differences between Groups 4 and 6 were not statistically significant. Thus, DCSA was not likely to be a sufficient discriminator of various clinical manifestations. In contrast, all parameters of MS-IRCE, WR-IRCE, MS/P1, WR/P1, MS/P2, and WR/P2 increased significantly as the group number decreased (*p* = 0.000). F values increased in the order of MS-IRCE, WR-IRCE, WR/P1, MS/P1, WR/P2 and MS/P2. In post hoc analysis, intergroup differences were evident in the order of MS-IRCE, MS/P1, WR/P1, MS/P2, WR-IRCE and WR/P2. These results showed that the order of the group was concordant with the degree of IRCE, which indicated intraradicular venous congestion or edema. In the diagnosis of lumbar spinal stenosis, grading with enhancement parameters (MS/P1 and WR/P1) could be feasible. In comparison to DCSA, these parameters revealed the superiority of segmentation in patients with lumbar central stenosis. Demographic data on sex and level distribution were not statistically different between groups (Table 2). The incidence of DVCE, RNR, NRSS and operation tended to decrease according to the order of group. These findings indicate that group order was significantly related to not only the severity of symptoms, but also related radiologic findings and prognosis. Stratification based on symptoms and radiologic findings can be applied in patients with central stenosis.

### 3.3. Correlation between DCSA/VAS and IRCE Parameters

DCSA significantly correlated with MS-IRCE (r = −0.548), WR-IRCE (r = −0.594), MS/P1 (r = −0.501), WR/P1 (r = −0.545), MS/P2 (r = −0.486), WR/P2 (r = −0.532), VAS leg (r = −0.306) and age (r = −0.172). VAS leg had a significant correlation with MS-IRCE (r = 0.489), WR-IRCE (r = 0.491), MS/P1 (r = 0.464), WR/P1 (r = 0.470), MS/P2 (r = 0.452) and WR/P2 (r = 0.457). As such, changes in DCSA and VAS leg could be explained by corresponding changes in enhancement parameters.

### 3.4. Clinical Interpretations of PVCE, DVCE and IRCE

The PVCE(+) group had significantly higher incidences of symptoms (95.1% vs. 45.2%), unilateral (37.7% vs. 1.1%) and operation (47.0% vs. 29.4%) than those in the PVCE(-) group (Table 3). The DVCE(+) group had significantly higher incidences of symptoms (91.1% vs. 32.1%), unilateral (29.9% vs. 16.1%), bilateral (58.4% vs. 14.3%) and operation (49.4% vs. 5.9%) compared with those in the DVCE(−) group. The IRCE(+) group had significantly higher incidences of symptoms (100% vs. 64.6%), bilateral (85.5% vs. 20.2%) and higher operation rate (70.3% vs. 17.9%) than those in the IRCE(−) group. The DVCE + PVCE(+) group had significantly higher incidences of symptoms (95.2% vs. 50.9%), unilateral (35.9% vs. 9.4%), bilateral (55.8% vs. 40.6%) and operation (49.5% vs. 26.0%) than those in the DVCE + PVCE(−) group. The DVCE + PVCE + IRCE(+) group had significantly higher incidences of symptoms (100% vs. 70.0%), bilateral (82.7% vs. 31.9%) and higher operation rate (67.5% vs. 26.3%) than those in the DVCE + PVCE + IRCE(−) group.

DVCE had the highest sensitivity, while DVCE + PVCE + IRCE or IRCE had the highest specificity in the prediction of symptom. For unilateral pain, PVCE indicated the highest sensitivity and DVCE + PVCE had the highest specificity. For operation, DVCE had the highest sensitivity and DVCE + PVCE + IRCE had the highest specificity. In the prediction of bilateral pain, DVCE had the highest sensitivity and DVCE + PVCE + IRCE had the highest specificity.

The presence of visible IRCE on sagittal or axial MR images revealed significantly higher VAS (7.3 vs. 4.5), lower DCSA (29.8 vs. 46.6 mm^2^) and higher enhancement parameters (MS-IRCE (369.8 vs. 257.1), WR-IRCE (307.3 vs. 223.2), MS/P1 (206.1 vs. 141.4), WR/P1 (171.1 vs. 122.8), MS/P2 (197.2 vs. 138.2), WR/P2 (163.7 vs. 120.0)) compared with those in the IRCE(−) group (*p* = 0.000).

### 3.5. Clinical Relationship and Enhancement Patterns of SPL, RNR, NRSS

SPL was significantly related to higher enhancement parameters (MS-IRCE (325.7 vs. 305.7, *p* = 0.023), WR-IRCE (272.6 vs. 260.0, *p* = 0.058), MS/P1 (180.8 vs. 169.3, *p* = 0.031), MS/P2 (174.3 vs. 163.7, *p* = 0.029)) and a higher incidence of IRCE (57.1% vs. 43.8%) and DVCE (90.9% vs. 81.2%) than those of the SPL(−) group (Table 4).

RNR was significantly related to higher enhancement parameters (MS-IRCE (339.6 vs. 288.3, *p* = 0.000), WR-IRCE (286.1 vs. 245.4, *p* = 0.000), MS/P1 (188.6 vs. 159.4, *p* = 0.000), WR/P1 (158.7 vs. 135.6, *p* = 0.000), MS/P2 (181.6 vs. 154.4, *p* = 0.000), WR/P2 (152.7 vs. 134.4, *p* = 0.000)), higher VAS leg (6.4 vs. 4.8, *p* = 0.000), lower DCSA (33.6 vs. 42.5 mm^2^, *p* = 0.000), higher operation rate (55.0% vs. 31.4%), higher incidence of symptoms (90.3% vs. 74.6%), IRCE (73.6% vs. 26.9%) and DVCE (91.0% vs. 77.7%) than those of the RNR(−) group.

NRSS had higher enhancement parameters (MS-IRCE (324.2 vs. 265.1, *p* = 0.000), WR-IRCE (274.0 vs. 226.9, *p* = 0.000), MS/P1 (180.0 vs. 145.8, *p* = 0.000), WR/P1 (151.9 vs. 124.7, *p* = 0.000), MS/P2 (173.4 vs. 142.2, *p* = 0.000), WR/P2 (146.4 vs. 121.7, *p* = 0.000)) and lower DCSA (35.6 vs. 48.7 mm^2^, *p* = 0.000) compared with the NRSS(−) group. Additionally, NRSS was significantly related to a higher operation rate (51.1% vs. 11.3%) and higher incidence of IRCE (58.0% vs. 11.3%) than those of NRSS(−) group.

### 3.6. Cutoff Values for the Prediction of Symptom, Bilaterality, Operation and Visible IRCE

In the analysis with all predictable clinical factors, enhancement parameters had a larger area under the curve in comparison with DCSA (Figure 8). Cutoff values for the prediction of symptoms were DCSA (41.8 mm^2^), MS-IRCE (269.5), WR-IRCE (230.5), MS/P1 (146.5), WR/P1 (127.7), MS/P2 (141.5) and WR/P2 (126.1) (Table 5). Cutoff values for bilateral leg pain were DCSA (41.1 mm^2^), MS-IRCE (289.5), WR-IRCE (249.5), MS/P1 (162.2), WR/P1 (138.6), MS/P2 (156.8) and WR/P2 (137.3). Cutoff values for decision-making on an operation were DCSA (35.5 mm^2^), MS-IRCE (294.5), WR-IRCE (253.5) MS/P1 (162.2) WR/P1 (139.2) MS/P2 (157.1) and WR/P2 (139.3). Cutoff values for visible IRCE on axial or sagittal images were DCSA (34.3 mm^2^), MS-IRCE (301.5), WR-IRCE (266.5), MS/P1 (168.3), WR/P1 (143.6), MS/P2 (161.3) and WR/P2 (139.5).

### 3.7. Illustrative Cases

Case 1. Patient (male, 52 years) had complained of bilateral buttock and posterior leg pain for three months. Neurogenic intermittent claudication was evident after 5 min of walking. This case demonstrated the selection of symptomatic level and the reversibilty of IRCE after epidural steroid injection (Figure 9).

Case 2. Patient (female, 56 years) had complained of bilateral (left > right) buttock and posterior leg pain for 2 months. This case showed the reversibilty of IRCE after decompressive surgeries (Figure 10).

Case 3. Patient (female, 74 years) had complained of pain in bilateral lower buttock and posterolateral thigh for 2 years. The pain was aggravating for 3 months. This case indicated the sequential treatment algorithm performing interventional procedures and decompressive surgeries in the multi-level lumbar stenosis (Figure 11).

## 4. Discussion

Conventional T1- or T2-weighted imaging does not provide excellent soft-tissue contrast for regions such as the intraspinal venous plexus. Gd enhancement on T1-weighted images with fat suppression could be considered to investigate structural information and the degree of venous congestion. It is more consistently and accurately depicted after the administration of gadolinium-diethylenetriamine pentaacetic acid (DTPA), which produces a uniformly high signal intensity of the epidural venous structures outside the extradural space. However, this technique has many limitations. The gray scale also has an impact on signal intensity displayed on the images since image values are not absolute. All values present in an MR image are distributed over the dynamic gray scale [9]. If the image is a T1-weighted image, the fat signal will occupy the highest values and thus will have the highest signal intensity on the image. When the fat signal is eliminated, the upper values of the dynamic gray scale represent non-fatty tissue [9]. In other words, the signal intensity of a certain tissue type, depends on other tissue types (included or excluded) in the image [9]. As such, the signal intensity of Gd does not have a linear relationship with the absolute concentration [10]. Thus, information on enhancement is qualitative. In clinical practice, the evaluation of enhancement effects is based on a visual inspection from each interpreter. Additionally, variation between MR scanners could be an obstacle to the absolute quantification of enhancement. The SPIR technique used in the present study suffers from tissue-related susceptibility and dielectric effects that could cause inhomogeneity in the Bo and B1 fields. This is manifested by incomplete fat suppression in certain regions of the image [11].

Several methods were applied to obtain the quantitative measurement of enhancement and to reduce all the limitations mentioned above. All images were acquired by the same protocol, platform and equipment in the same hospital. Instead of absolute measurement, a signal ratio between rootlets and psoas muscle was chosen. To avoid partial volume effects, the margin of rootlets was manually delineated on the T2-weighted axial image and subsequently, the contour was transferred into the corresponding location on the T1-weighted axial image with fat suppression. On the same image, the signal intensity of the psoas muscle was measured in the same way to prevent the effect of the dynamic gray scale. However, it was found that signal inhomogeneity, even within the psoas muscle, was changeable according to the anteroposterior direction and distance from the midline. In addition, restricted ROI as Psoas2 was attempted to compare the superiority of methodology, but showed higher signal inhomogeneity than Psoas1, depending on the sites. Nonetheless, IRCE parameters were confirmed as a severity evaluation index in the present study.

Currently, there is no consensus on the specific diagnostic criteria for degenerative LSS. A reduced DCSA is a good discriminator for the presence of LSS [12,13,14]. However, the use of cross-sectional area (CSA) as a diagnostic tool is not ideal. Such criteria can lead to both over- and under-diagnosis of clinically relevant LSS [15] as it is known that CSA often correlates poorly with clinical symptoms in LSS [16]. Patients with RNRs are older, exhibit a long period between the onset of the symptoms and the time of MRI and experience more symptoms [17,18]. Patients without RNRs had a slightly better (but not statistically significant) surgical outcome than patients with RNRs [19]. NRSS has been described as a new diagnostic test for lumbar spinal stenosis [12]. However, the absence of NRSS could not aid in the differential diagnosis between LSS and low back pain or vascular claudication or add any specific diagnostic information beyond the traditional history, physical examination and imaging studies [20,21]. This absence of NRSS was associated with a small, but significantly greater surgical treatment effect on ODI compared to the presence of NRSS [20]. In clinical settings, such radiological findings cannot grade the severity of symptoms. This could be because the real pathoanatomical process in the rootlets was not reflected. The pathomechanism in lumbar canal stenosis was related to intraradicular venous congestion due to the occlusion of subarachnoid space [1]. In addition to experimental results of Kobayashi et al. [1], the author assumed that extradural venous plexus (IVVP) could be congested at an early stage and venous hypertension developed due to reflux from IVVP to the intradural radicular veins could play a crucial role at a late stage. This is because the radicular veins regulate venous reflux in the absence of a valve. Diverse symptoms—including deep muscular pain, weakness and loss of sensation in the leg—could be due to venous congestion in intrathecal rootlets. Differential intraradicular pathologic changes were expected in patients with such symptoms. Accordingly, the quantitative measurement of congestion at IVVP and intraradicular vessels could be helpful in the prediction of LSS symptoms.

In this study, grouping was conducted on the basis of the expected distribution of venous congestion. The group number indicated the order of severity in intraradicular venous congestion or edema. Although intergroup differences among Groups 3, 4 and 5 were not statistically significant, each group had differential enhancement parameters. This finding could be due to the small number of patients in Group 4. In a large-scale study with sufficient patients, statistical significance could be anticipated. According to results obtained in the present study, stratification or grading by MS/P1 and WR/P1 could be more useful in comparison to DCSA.

Pain distribution alone could only provide limited information. Not all patients with lumbar central stenosis present with a typical symptom like neurogenic claudication. There is no correlation between the severity of stenosis and clinical complaint. The radiologic findings such as DCSA, SPL, NRSS and RNR may correspond to environmental and indirect causes for developing a root injury. They could not show the final outcome like a pathologic process that affects nerves at the root level (radiculopathy). There are certainly limits to correlation between radiologic findings and clinical complaints. As clinical symptoms are related to the neuropathological change (axonal damage) in the root, electrodiagnostic studies, especially electromyography (EMG), can provide confirmatory information. The primary use of EMG is to confirm nerve root dysfunction when the diagnosis is uncertain or to distinguish a radiculopathy from other lesions when the physical examination findings are unclear. Although EMG is very sensitive and specific, normal EMG results in a patient with signs and symptoms consistent with a radiculopathy do not exclude the diagnosis of radiculopathy. For example, a pure sensory lesion at the root level will have negative EMG and sensory nerve conduction study results. In this study, IRCE is also a kind of radiologic finding. Nevertheless, it has a pathologic meaning of venous hypertension which induces axonal injuries. The validity of IRCE may be considered as somewhere between DCSA and EMG. Additionally, IRCE includes changes from all sensorimotor rootlets and has also the virtue of simplicity and good visualization compared with EMG in the selection of symptomatic level in lumbar central stenosis. These advantages will be able to benefit clinicians immediately for performing pain interventions and planning a minimally invasive surgery. Considering many aspects, IRCE may be complementary to neurologic examination, DCSA and EMG results in patients with less obvious clinical complaints. So, patients with bilateral symptom were divided into IRCE(+) and IRCE(−) groups. Between Groups 1 and 3, there was no statistical difference in VAS leg (7.4 ± 1.3 vs. 7.0 ± 1.4); however, the operation rate (74% vs. 46%) was significantly different. The incidence of SPL (26% vs. 17%), RNR (65% vs. 44%) and NRSS (94% vs. 81%) was higher in Group 1 than Group 3. In addition, patients with unilateral symptom were divided into IRCE(+) and IRCE(−). Between Groups 2 and 5, there was no statistical difference in VAS leg (7.0 ± 1.1 vs. 7.0 ± 1.7), however, the operation rate (47% vs. 21%) was significantly different. The incidence of SPL (39% vs. 19%), RNR (74% vs. 10%) and NRSS (96% vs. 36%) were statistically higher in Group 2 than Group 5. Collectively, the pain score is not likely to reflect enhancement parameters that could indicate intraradicular pathologic changes. Most likely, there is a ceiling effect of VAS over the pathologic threshold of intraradicular venous congestion. However, the higher incidence of radiological findings (SPL, RNR, NRSS) in the IRCE(+) group, compared with the IRCE(−) group, supports intergroup differences in both pathologic processes, which indicated the intraradicular congestion and prognosis and was reflected by the operation rate. Stratification should be designed meticulously under the consideration of both pathologic and symptomatic points.

A gradual extension of venous congestion among enhancement patterns was expected. In this study, analyses were performed in the following manner. PVCE was symbolic of unilateral leg pain (sensitivity 99%, specificity 38%, accuracy 55%), DVCE signified the development of symptoms (93%, 60%, 87%), PVCE + DVCE represented the development of symptoms (80%, 83%, 81%), and IRCE denoted a bilateral symptom (79%, 86%, 82%). The order of appearance in PVCE, DVCE, and IRCE may be concordant with the severity of venous congestion. Extrathecal events are likely to play a major role in the pathogenesis of lumbar central stenosis compared with intrathecal events.

SPL showed a significant relationship with the development of enhancement patterns, but not with clinical manifestations. The absence of NRSS had the higher incidence of IRCE and operation than the NRSS(+) group. RNR was related to the higher incidence of symptom, operation, DVCE and IRCE than those of the RNR(−) group. All radiologic findings (SPL/NRSS/RNR) were significantly associated with the change of enhancement patterns. However, enhancement patterns were likely to be superior to abovementioned radiologic parameters in the prediction of clinical manifestations.

The cutoff values of enhancement parameters for the prediction of clinical factors such as the development of symptom, bilateral leg pain, operation, and visible IRCE on the axial or sagittal images were estimated. These values increased sequentially in the order of the abovementioned clinical factors and could be in concordance with the severity of congestion. With these benchmarks, it is possible to predict when symptoms could be revealed; to explain whether a severe pathologic change (visible IRCE) exists; and to recommend at what of venous congestion an operation is necessary.

A potential limitation of the study could be the accuracy in measurement of signal intensities. In conventional MR systems, the patient is scanned in the supine position. Venous congestion as the cause of neurogenic claudication cannot be fully evaluated. Dynamic compression during walking or repetitive exertional events cannot be examined. Nevertheless, it seems that venous enhancement patterns (PVCE, DVCE, IRCE) can be shown at rest on the fat-suppressed contrast-enhanced T1-weighted images. This may be because static compression by circumjacent bone and soft tissue is enough to develop venous hypertension. This means that venous hypertension and neuronal damage are in an advanced stage. From the physiological point of view, upright and post-exertional MRI will be the best method to evaluate the venous congestion in incipient stage (not visible on supine MRI, but visible on upright MRI). Unfortunately, there are some compromises until now. While traditional supine scanners have a high strength magnetic field (generally 1.5 T, but sometimes up to 3 T), upright and open MRI scanners have a lower field strength (0.2–0.6 T) with some compromise in image quality. Moreover, fat-suppressed contrast-enhanced T1-weighted images may have a problem relating to field inhomogeneity due to this field strength. The constant improvement in both coil technology and software will have narrowed the gap in image quality between low and higher field systems. Further studies using novel MR techniques with less field inhomogeneity are needed to reduce the variance of signal intensity. Another limitation is that the acquisition of statistical significance failed due to low case incidence among Groups 2, 3 and 4. However, intergroup differences were already confirmed between Group 2 and Group 3. Intergroup differences from group 3 to group 5 will be determined if more cases are collected in the prospective study.

To conclude, with enhancement patterns and parameters, stratification and grading are feasible in patients with lumbar central stenosis. Cutoff values could help in the prediction of symptoms and decision-making at the surgical level. Collectively, such information could enable target-oriented interventional procedures or minimally invasive surgeries in patients with multilevel stenosis.

## Figures and Tables

**Figure 1 jcm-09-03084-f001:**
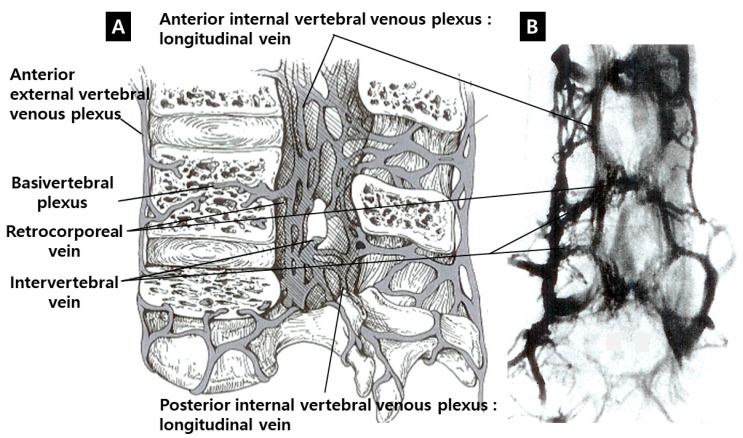
Three-dimensional structures of internal and external vertebral venous plexus. (**A**) The vertebral venous plexus was divided into three intercommunicating divisions: internal vertebral, external vertebral and basivertebral plexus; (**B**) venography on a coronal plane. (Reproduced with permission from Gregory D. Cramer and Susan A. Darby, Clinical Anatomy of the Spine, Spinal cord and ANS 3rd ed.; published by Elsevier, 2014).

**Figure 2 jcm-09-03084-f002:**
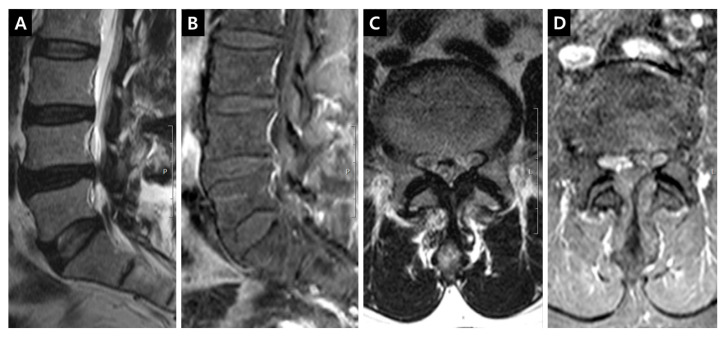
Enhancement patterns: periradicular vein (PVCE). (**A**) T2-weighted right parasagittal image showing the origin of right L5 traversing root; (**B**) Fat-suppressed enhanced T1-weighted right parasagittal image indicating the periannular enhancement on the right subarticular L4-5 annulus; (**C**) T2-weighted axial image on the L5 upper endplate level; (**D**) Fat-suppressed enhanced T1-weighted axial image on the L5 upper endplate level, the definite periradicular enhancement around the right L5 traversing root can be seen in the lateral recess.

**Figure 3 jcm-09-03084-f003:**
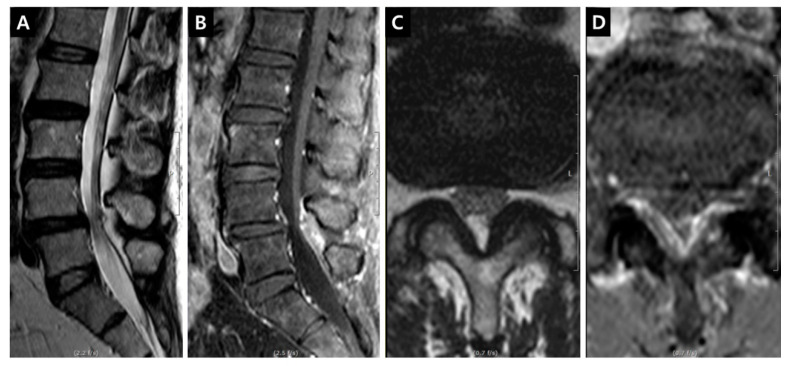
Enhancement patterns: dorsal epidural vein (DVCE). (**A**) T2-weighted midsagittal image; (**B**) Fat-suppressed enhanced T1-weighted midsagittal image showing the prominent enhancement on the dorsal dural surface; (**C**) T2-weighted axial image on the L4-5 disc level; (**D**) Fat-suppressed enhanced T1-weighted axial image on the L4-5 disc level, the definite enhancement on the ventral surface of ligamentum flavum in the dorsal epidural space can be seen.

**Figure 4 jcm-09-03084-f004:**
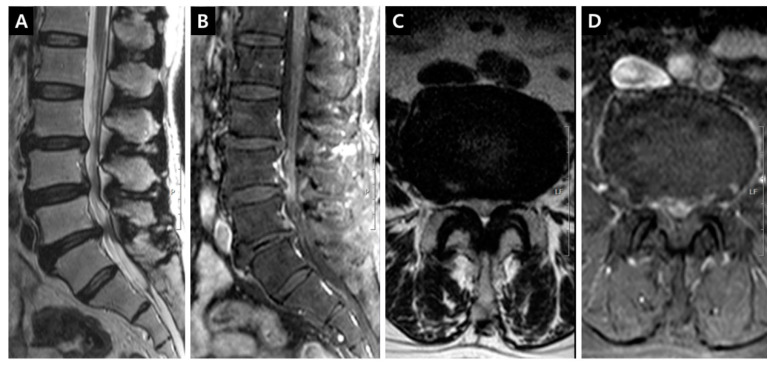
Enhancement patterns: intraradicular vein of cauda equina (IRCE). (**A**) T2-weighted midsagittal image showing L3-4 central protrusion, L4-5 central stenosis, and L5-6 central protrusion; (**B**) Fat-suppressed enhanced T1-weighted midsagittal image revealing the prominent enhancement of intradural rootlets on the L4-5 disc level; (**C**) T2-weighted axial image on the L4-5 disc level; (**D**) Fat-suppressed enhanced T1-weighted axial image on the L4-5 disc level, the definite enhancement of rootlets in the thecal sac can be seen.

**Figure 5 jcm-09-03084-f005:**
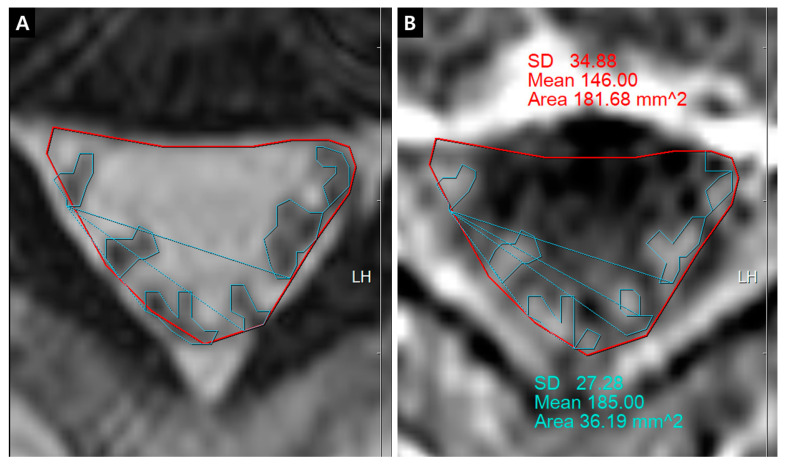
Whole rootlets IRCE (WR-IRCE) on L4–5 as a control. (**A**) T2-weighted axial image; (**B**) fat-suppressed enhanced T1-weighted axial image. SD—standard deviation

**Figure 6 jcm-09-03084-f006:**
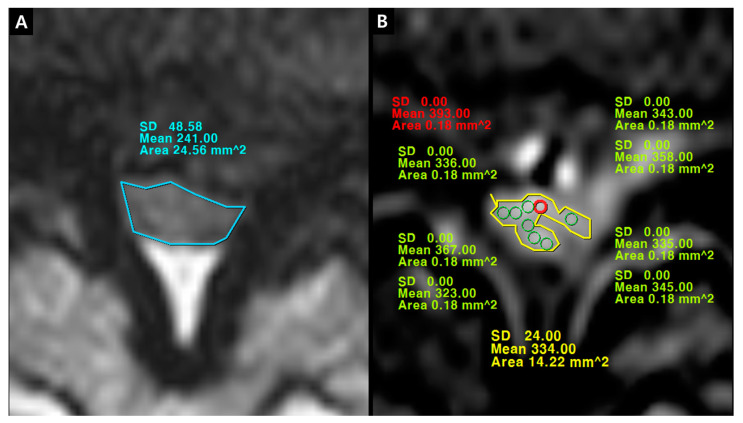
(**A**) dural cross sectional area (DCSA) on T2-weighted axial image; (**B**) whole rootlets IRCE (WR-IRCE, yellow color) and maximal spot rootlet IRCE (MS-IRCE, red color) on fat-suppressed enhanced T1-weighted axial image among signal intensities of manually-defined whole rootlets and each rootlet.

**Figure 7 jcm-09-03084-f007:**
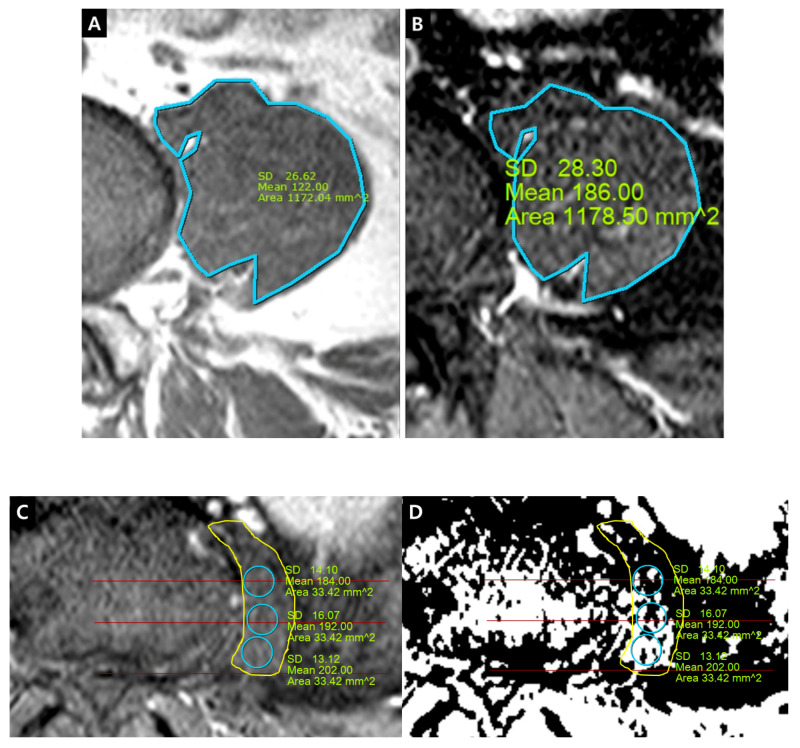
Signal intensity measurement of psoas muscle on fat-suppressed enhanced T1-weighted axial image. (**A**,**B**) Psoas1, manually-defined region of interests (ROIs); (**C**,**D**) Psoas2, circular ROIs.

**Figure 8 jcm-09-03084-f008:**
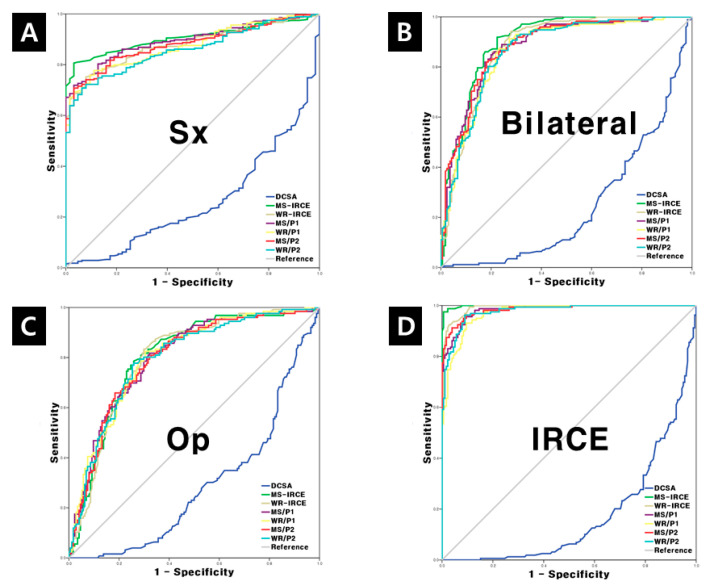
Receiver-operator characteristic (ROC) curve and cutoff values for the prediction of (**A**) symptom, (**B**) bilateral leg pain, (**C**) determination of operation and (**D**) visible IRCE on fat-suppressed enhanced T1-weighted axial image. Sx,—symptom, Op—operation

**Figure 9 jcm-09-03084-f009:**
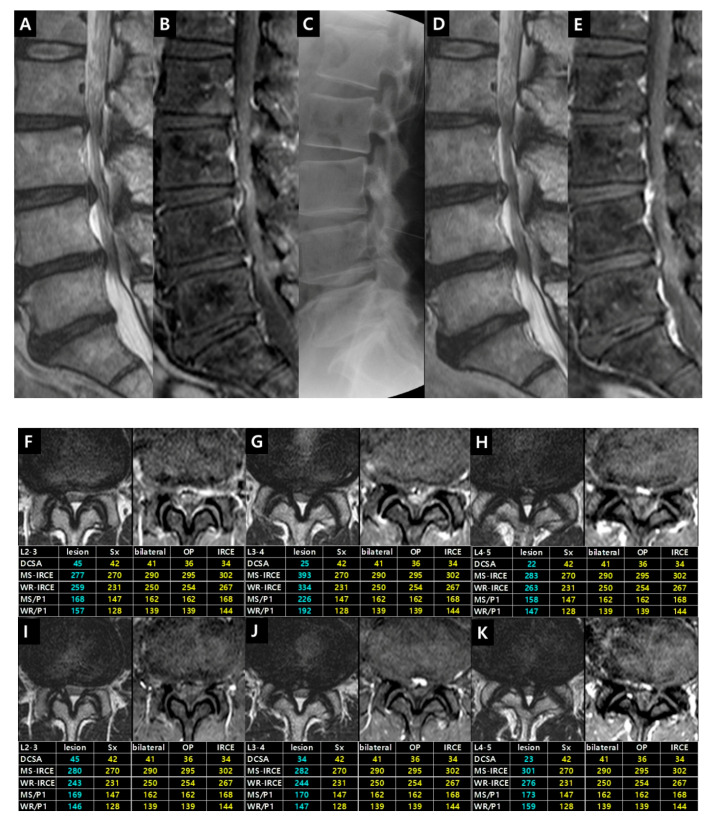
Illustrative Case 1. (**A**) T2-weighted and (**B**) fat-suppressed enhanced T1-weighted sagittal images showed L2–3, L3–4 and L4–5 central stenosis. (**C**) An interlaminar epidural steroid injection was performed at L3–4 level. The pain has disappeared for two years. Follow-up MR imaging was taken due to the new development of paresthesia in the left lateral thigh. Follow-up T2-weighted (**D**) and fat-suppressed enhanced T1-weighted (**E**) sagittal images demonstrated the disappearance of IRCE on L3–4 level and the appearance of IRCE on L4–5 level. The disappearance of pain and IRCE on L3–4 level after intervention is likely to be caused by a partial absorption of the protruded central disc (the increase of DCSA). T2-weighted and fat-suppressed enhanced T1-weighted axial images on L2–3 (**F**), L3–4 (**G**), and L4–5 (**H**) illustrated the definite visible IRCE with higher IRCE parameters on L3–4 level than L2–3 and L4–5 level. Follow-up images on L2–3 (**I**), L3–4 (**J**), and L4–5 (**K**) revealed no change on L2–3 level, the disappearance of IRCE on L3–4 level, and the increased IRCE parameters on L4–5 level compared with previous images. Changes in enhancement parameters could explain that the symptomatic level changed from L3–4 to L4–5 level. Epidural steroid injection was performed on L4–5 level. The patient has improved.

**Figure 10 jcm-09-03084-f010:**
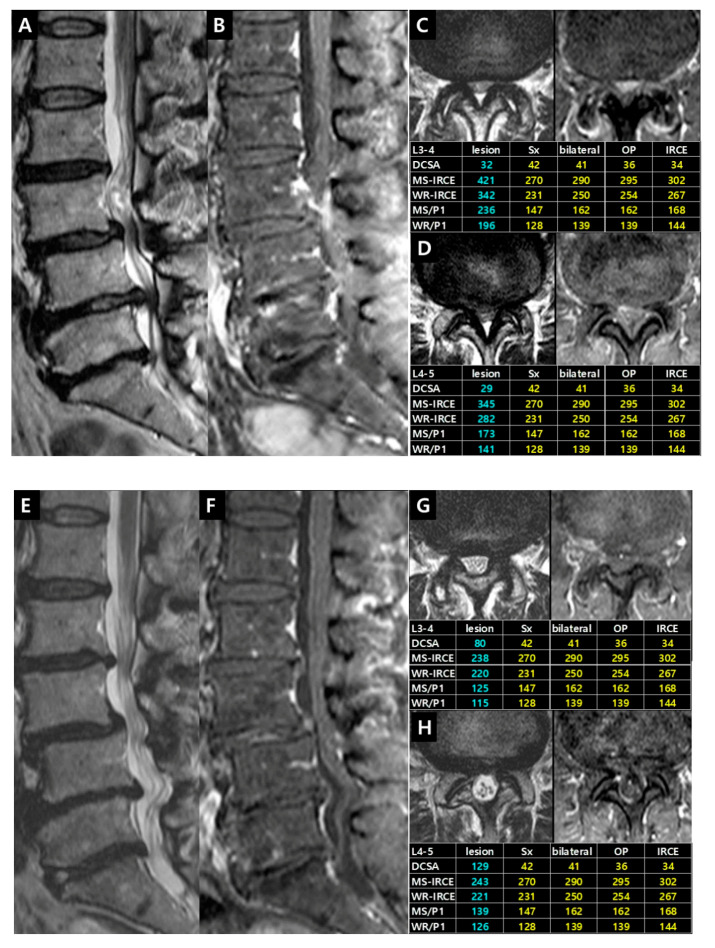
Illustrative Case 2. (**A**,**B**) Sagittal MR images revealed the definite central stenosis on L3–4 and L4–5 level; (**C**,**D**) Axial images showed IRCEs and the increased enhancement parameters on both levels. For dual lesions, minimally invasive L3 and L4 dome laminoplasties were performed and symptoms improved significantly; (**E**,**F**) Five years later, IRCEs on L3–4 and L4–5 level disappeared on follow-up sagittal images; (**G**,**H**) The decrease of enhancement parameters corresponded well with all clinical and radiological changes on follow-up axial images.

**Figure 11 jcm-09-03084-f011:**
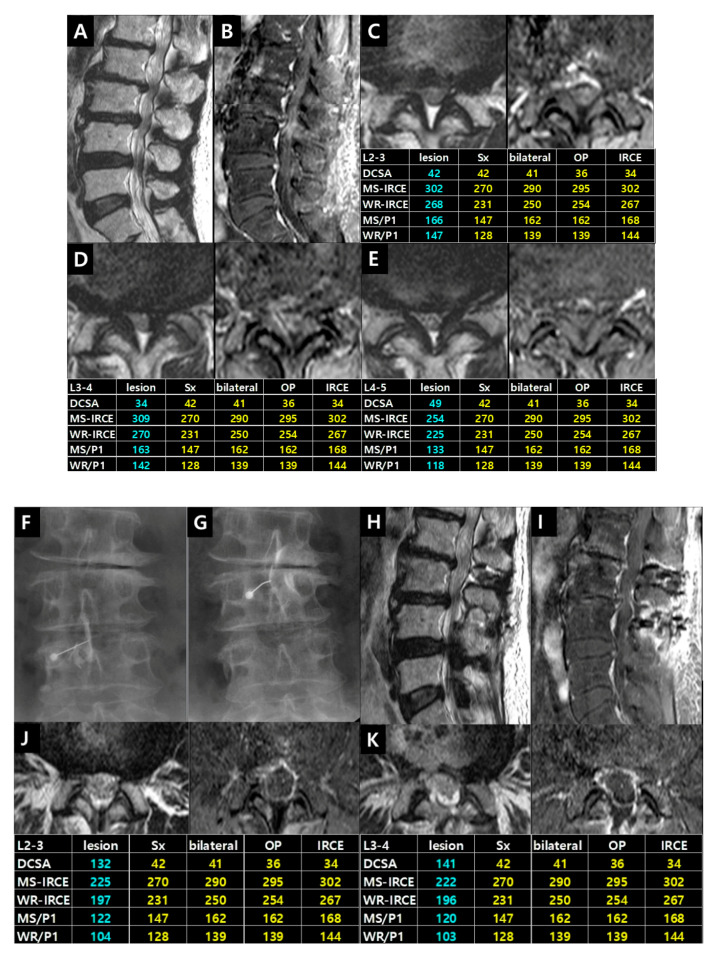
Illustrative Case 3. (**A**,**B**) T2-weighted and enhanced fat-suppressed T1-weighted sagittal images showed central stenosis on the L2–3, L3–4 and L4–5 level. Axial images revealed higher signals and enhancement parameters on (**C**) L2–3 and (**D**) L3–4 level than (**E**) L4–5 level. The interlaminar injection was performed on (**F**) L3–4 level at first. Symptoms improved remarkably; (**G**) Subsequent injection on L2–3 level also improved the pain. (**H**,**I**) After L2 and L3 dome laminoplasties, all symptoms disappeared. One week later, follow-up axial images demonstrated the expansion of the thecal sac and the disappearance of IRCEs on (**J**) L2–3 and (**K**) L3–4 level. The decrease of enhancement parameters indicated the immediate improvement after surgery.

**Table 1 jcm-09-03084-t001:** Comparison of age, VAS, DCSA, MS-IRCE, WR-IRCE, MS/P1, WR/P1, MS/P2, WR/P2.

Groups	Age	VAS	DCSA (mm^2^)	MS-IRCE	WR-IRCE	Psoas1	MS/P1 (%)	WR/P1 (%)	Psoas2	MS/P2 (%)	WR/P2 (%)
IRCE(+), bilateral	68.0 ± 9.7 ^a,b,c,d^	7.4 ± 1.3 ^a^	30.3 ± 10.4 ^a^	371.7 ± 47.5 ^a^	307.9 ± 35.2 ^a^	180.6 ± 16.7	207.6 ± 33.2 ^a^	171.7 ± 23.6 ^a^	188.6 ± 16.4	198.5 ± 30.9 ^a^	164.2 ± 22.0 ^a^
*n* = 136											
IRCE(+), unilateral	69.1 ± 8.9 ^a,b,c,d,e^	7.0 ± 1.1 ^a^	26.6 ± 8.8 ^a^	350.0 ± 47.1 ^a^	304.2 ± 26.7 ^a^	181.9 ± 10.8	197.6 ± 24.3 ^a^	167.5 ± 14.6 ^a^	189.0 ± 6.9	189.8 ± 22.9 ^a^	160.9 ± 12.4 ^a^
*n* = 23											
IRCE(−), bilateral	70.5 ± 8.2 ^a,b,c,d^	7.0 ± 1.4 ^a^	38.9 ± 13.7 ^b,c^	283.2 ± 12.5 ^b^	244.1 ± 14.4 ^b,c^	183.3 ± 17.9	155.9 ± 15.8 ^b,c^	134.4 ± 14.6 ^b,c^	188.4 ± 14.0	151.1 ± 12.1 ^b,c^	130.2 ± 11.0 ^b,c^
*n* = 36											
IRCE(−), midsacral	58.8 ± 12.0 ^a,b,c,d,e^	6.0 ± 2.1 ^a^	45.5 ± 10.3 ^b,c,d^	268.6 ± 7.6 ^c^	231.0 ± 12.2 ^b,c,d^	180.1 ± 6.3	149.2 ± 5.0 ^b,c,d^	128.4 ± 9.1 ^b,c,d^	188.2 ± 5.8	142.8 ± 6.0 ^b,c,d^	122.8 ± 7.7 ^b,c,d^
*n* = 9											
IRCE(−), unilateral	63.6 ± 9.4 ^b,d,e^	7.0 ± 1.7 ^a^	48.9 ± 15.3 ^c,d^	255.9 ± 27.2 ^d^	221.0 ± 26.1 ^c,d,e^	180.9 ± 9.3	141.8 ± 17.0 ^c,d^	122.8 ± 15.6 ^c,d^	185.9 ± 5.6	137.7 ± 15.2 ^c,d,e^	119.3 ± 14.0 ^c,d^
*n* = 70											
IRCE(−), no Sx	68.2 ± 9.1 ^a,b,c,d,e^	0	48.5 ± 13.1 ^c,d^	241.8 ± 18.6 ^e^	212.0 ± 19.5 ^d,e^	184.6 ± 14.3	131.6 ± 12.5 ^e^	115.3 ± 11.7 ^e^	185.6 ± 13.1	130.8 ± 12.2 ^d,e^	114.6 ± 11.3 ^c,d^
*n* = 63											
Control	68.1 ± 5.5 ^a,b,c,d,e^	0	133.5 ± 40.7 ^e^	233.6 ± 15.6 ^e^	199.6 ± 13.1 ^f^	188.9 ± 9.4	123.9 ± 10.8 ^f^	105.9 ± 8.6 ^f^	189.1 ± 7.8	123.8 ± 9.7 ^f^	105.7 ± 17.7 ^e^
*n* = 30, L3–4 or L4–5											
F-value	4.335	2.411	35.149	174.347	164.015	1.847	127.309	129.125	0.754	122.066	123.978
*p* value *	0	0.05	0	0	0	0.089	0	0	0.606	0	0

*—statistical significances tested by one-way analysis of variances among groups. ^a,b,c,d,e,f^—same letters indicate statistically insignificant difference between groups based on Games-Howell’s multiple comparison test (post hoc analysis); VAS—visual analogue scale; IRCE—intraradicular enhancement; Sx—symptom; DCSA—dural cross sectional area; MS-IRCE—maximal spot rootlet IRCE; WR-IRCE—whole rootlets IRCE; MS/P1—(MS-IRCE/Psoas1)*100; WR/P1—(WR-IRCE/Psoas1)*100; MS/P2—(MS-IRCE/Psoas2*100); WR/P2—(WR-IRCE/Psoas2)*100.

**Table 2 jcm-09-03084-t002:** Intergroup differences of categorical variables.

Groups (*n*, %)	Female	L3–4 + L4–5	DVCE	PVCE	PVCE + DVCE	PVCE + DVCE + IRCE	SPL	RNR	NRSS	OP *	Outcome *
IRCE(+), bilateral	81, 60%	125, 92%	129, 95%	106, 78%	105, 77%	105, 77%	35, 26%	89, 65%	128, 94%	88, 74%	112, 94%
*n* = 136 (F/U loss 17)											
IRCE(+), unilateral	16, 70%	22, 96%	23, 100%	22, 96%	22, 96%	22, 96%	9, 39%	17, 74%	22, 96%	9, 47%	19, 100%
*n* = 23 (F/U loss 4)											
IRCE(−), bilateral	20, 56%	31, 86%	35, 97%	25, 69%	24, 67%	0, 0%	6, 17%	16, 44%	29, 81%	15, 46%	26, 85%
*n* = 36 (F/U loss 3)											
IRCE(−), midsacral	5, 56%	9, 100%	8, 89%	9, 100%	8, 89%	0, 0%	5, 56%	1, 11%	6, 67%	2, 22%	9, 100%
*n* = 9 (F/U loss 0)											
IRCE(−), unilateral	42, 60%	63, 90%	61, 87%	70, 100%	61, 87%	0, 0%	13, 19%	7, 10%	25, 36%	12, 21%	53, 93%
*n* = 70 (F/U loss 13)											
IRCE(−), no Sx	36, 57%	51, 81%	25, 40%	12, 19%	11, 18%	0, 0%	9, 14%	14, 22%	47, 75%	NA	NA
*n* = 63											
Control	16, 53%	30, 100%	0, 0%	0, 0%	0, 0%	0, 0%	0, 0%	0, 0%	0, 0%	NA	NA
*n* = 30, L3–4 or L4–5											

*—favorable outcome, calculated in the levels with a final follow-up possible; IRCE—intraradicular enhancement; Sx—symptom; DVCE—dorsal epidural vein enhancement; PVCE—periradicular enhancement; SPL—spondylolisthesis; RNR—redundant nerve root; NRSS—nerve root sedimentation sign; OP—operation; outcome—excellent + good.

**Table 3 jcm-09-03084-t003:** Clinical meaning of PVCE, DVCE and IRCE.

Factor vs. Response	*p* Value	Sensitivity	Specificity	Accuracy	PPV	NPV	OR	RR
PVCE vs. Sx	0.000	0.85	0.81	0.84	0.95	0.55	23.48	2.11
PVCE vs. Unilateral	0.000	0.99	0.38	0.55	0.38	0.99	55.68	35.07
PVCE vs. Operation	0.005	0.80	0.34	0.53	0.47	0.71	2.13	1.60
DVCE vs. Sx	0.000	0.93	0.60	0.87	0.91	0.68	21.62	2.83
DVCE vs. Unilateral	0.035	0.90	0.19	0.39	0.30	0.84	2.23	1.86
DVCE vs. Bilateral	0.000	0.95	0.29	0.63	0.58	0.86	8.41	4.09
DVCE vs. Operation	0.000	0.98	0.28	0.57	0.49	0.94	15.62	8.40
PVCE + DVCE vs. Sx	0.000	0.80	0.83	0.81	0.95	0.49	19.26	1.87
PVCE + DVCE vs. Unilateral	0.000	0.89	0.39	0.53	0.36	0.91	5.38	3.81
PVCE + DVCE vs. Bilateral	0.009	0.78	0.37	0.57	0.56	0.59	1.85	1.38
PVCE + DVCE vs. Operation	0.000	0.80	0.41	0.57	0.50	0.74	2.78	1.90
IRCE vs. Sx	0.000	0.58	1.00	0.66	1.00	0.35	∞	1.55
IRCE vs. Unilateral	0.000	0.25	0.44	0.39	0.14	0.61	0.26	0.37
IRCE vs. Bilateral	0.000	0.79	0.86	0.82	0.86	0.81	23.32	4.20
IRCE vs. Operation	0.000	0.77	0.76	0.77	0.70	0.82	10.85	3.93
PVCE + DVCE + IRCE vs. Sx	0.000	0.46	1.00	0.56	1.00	0.30	∞	1.43
PVCE + DVCE + IRCE vs. Unilateral	0.001	0.24	0.57	0.48	0.17	0.65	0.41	0.51
PVCE + DVCE + IRCE vs. Bilateral	0.000	0.61	0.87	0.74	0.83	0.68	10.19	2.59
PVCE + DVCE + IRCE vs. Operation	0.000	0.61	0.79	0.71	0.68	0.74	5.82	2.56

IRCE—intraradicular enhancement; DVCE—dorsal epidural enhancement; PVCE—periradicular enhancement; Sx—symptom; PPV—positive predictive value; NPV—negative predictive value; OR—odds ratio; RR—relative risk.

**Table 4 jcm-09-03084-t004:** Clinical relationship with enhancement patterns of SPL, RNR, NRSS.

Factor vs. Response	*p* Value	Sensitivity	Specificity	Accuracy	PPV	NPV	OR	RR
SPL vs. Sx	0.073	0.25	0.86	0.36	0.88	0.21	1.98	1.11
SPL vs. OP	0.172	0.26	0.80	0.58	0.49	0.60	1.46	1.23
SPL vs. DVCE	0.043	0.25	0.88	0.35	0.91	0.19	2.32	1.12
SPL vs. IRCE	0.046	0.28	0.81	0.56	0.57	0.56	1.68	1.29
SPL vs. RNR	0.417	0.25	0.79	0.56	0.47	0.56	1.24	1.13
SPL vs. NRSS	0.697	0.23	0.79	0.36	0.78	0.24	1.13	1.03
RNR vs. Sx	0.000	0.47	0.78	0.53	0.90	0.25	3.16	1.21
RNR vs. OP	0.000	0.58	0.66	0.63	0.55	0.67	2.67	1.75
RNR vs. DVCE	0.001	0.47	0.77	0.52	0.91	0.22	2.89	1.17
RNR vs. IRCE	0.000	0.67	0.79	0.73	0.74	0.73	7.37	2.68
RNR vs. NRSS	0.000	0.55	0.98	0.65	0.99	0.40	48.16	1.65
NRSS vs. Sx	0.732	0.77	0.25	0.67	0.82	0.20	1.12	1.02
NRSS vs. OP	0.000	0.93	0.36	0.60	0.51	0.87	7.20	4.03
NRSS vs. DVCE	0.352	0.77	0.29	0.69	0.84	0.20	1.36	1.06
NRSS vs. IRCE	0.000	0.94	0.40	0.66	0.58	0.89	11.06	5.19

IRCE—intraradicular enhancement, Sx—symptom, DVCE—dorsal epidural enhancement, PPV—positive predictive value, NPV—negative predictive value, OR—odds ratio; RR—relative risk; SPL—spondylolisthesis; RNR—redundant nerve root; NRSS—nerve root sedimentation sign; OP—operation.

**Table 5 jcm-09-03084-t005:** Cutoff values for the prediction of clinical factors.

Factor	Parameters	Area	*p* Value	Cutoff	Sensitivity (%)	Specificity (%)
**Sx (Y/N)**	DCSA (mm^2^)	0.738	0.000	41.8	70	69
	MS-IRCE	0.906	0.000	269.5	81	97
	WR-IRCE	0.879	0.000	230.5	79	84
	MS/P1 (%)	0.896	0.000	146.5	80	87
	WR/P1 (%)	0.876	0.000	127.7	78	89
	MS/P2 (%)	0.880	0.000	141.5	82	82
	WR/P2 (%)	0.860	0.000	126.1	75	87
**Bilateral**	DCSA (mm^2^)	0.754	0.000	41.1	81	60
	MS-IRCE	0.903	0.000	289.5	86	82
	WR-IRCE	0.881	0.000	249.5	86	79
	MS/P1 (%)	0.881	0.000	162.2	85	80
	WR/P1 (%)	0.866	0.000	138.6	86	78
	MS/P2 (%)	0.883	0.000	156.8	82	82
	WR/P2 (%)	0.867	0.000	137.3	80	81
**Op (Y/N)**	DCSA (mm^2^)	0.705	0.000	35.5	65	67
	MS-IRCE	0.797	0.000	294.5	79	72
	WR-IRCE	0.793	0.000	253.5	82	71
	MS/P1 (%)	0.796	0.000	162.2	80	69
	WR/P1 (%)	0.798	0.000	139.2	83	69
	MS/P2 (%)	0.790	0.000	157.1	76	70
	WR/P2 (%)	0.790	0.000	139.3	77	75
**IRCE (Y/N)**	DCSA (mm^2^)	0.824	0.000	34.3	67	80
	MS-IRCE	0.999	0.000	301.5	99	98
	WR-IRCE	0.991	0.000	266.5	93	97
	MS/P1 (%)	0.983	0.000	168.3	93	93
	WR/P1 (%)	0.972	0.000	143.6	93	90
	MS/P2 (%)	0.984	0.000	161.3	92	94
	WR/P2 (%)	0.977	0.000	139.5	93	92

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
