# Peer review of "Quantification of Lumbar Spinal Canal Stenosis by Quantitative Fat-Suppressed Contrast-Enhanced Magnetic Resonance Imaging"

_jcm, 2020, doi:10.3390/jcm9103084_

Round 1

Reviewer 1 Report

This paper is very interesting, correctly written, with an adequate satistical study, but its validity does not seem absolute to me, because, in my experience and in literature, mild-to- stenosis according to MR images with evident venous congestion, can also be found in asymptomatic individuals.  The best diagnostic test for the diagnosis of lumbar central stenosis ( LCS)  is magnetic resonance imaging (MRI) of course. For cases showing typical neurogenic claudication symptoms and unequivocal imaging findings, the diagnosis is straightforward. However, in my experiences  not all patients present with typical symptoms, and there is obviously no correlation between the severity of stenosis (computed tomography and MRI) and clinical complaint.

In these cases electrophysiological recordings are complementary to the neurologic examination and can provide confirmatory information in less obvious clinical complaints. These aspects should be developed in the discussion.

Author Response

Point 1: This paper is very interesting, correctly written, with an adequate statistical study, but its validity does not seem absolute to me, because, in my experience and in literature, mild-to- stenosis according to MR images with evident venous congestion, can also be found in asymptomatic individuals. 

Response 1: I agree with you. In my series, 63 asymptomatic patients had shown epidural enhancement patterns like DVCE (40%), PVCE (19%), and DVCE+PVCE (18%). However, IRCE pattern was not found. That was why IRCE had the highest specificity relating to the presence of symptom. According to enhancement patterns, different clinical meanings should be considered.

Point 2: The best diagnostic test for the diagnosis of lumbar central stenosis (LCS) is magnetic resonance imaging (MRI) of course. For cases showing typical neurogenic claudication symptoms and unequivocal imaging findings, the diagnosis is straightforward. However, in my experiences, not all patients present with typical symptoms, and there is obviously no correlation between the severity of stenosis (computed tomography and MRI) and clinical complaint. In these cases, electrophysiological recordings are complementary to the neurologic examination and can provide confirmatory information in less obvious clinical complaints. These aspects should be developed in the discussion.

Response 2: Thank you for commenting the weakness of previously-reported radiologic finding like DCSA and the relationship with electrodiagnostic studies. Totally, I agree with you. As you suggested, a paragraph given below was added in the discussion. In addition, I cordially ask your permission of quoting or paraphrasing several passages or sentences in your comments.

Not all patients with lumbar central stenosis present with a typical symptom like neurogenic claudication. There is no correlation between the severity of stenosis and clinical complaint. The radiologic findings such as DCSA, SPL, NRSS, and RNR may correspond to environmental and indirect causes for developing a root injury. They could not show the final outcome like a pathologic process that affects nerves at the root level (radiculopathy). There are certainly limits to correlation between radiologic findings and clinical complaints. As clinical symptoms are related to the neuropathological change (axonal damage) in the root, electrodiagnostic studies, especially electromyography (EMG), can provide confirmatory information. The primary use of EMG is to confirm nerve root dysfunction when the diagnosis is uncertain or to distinguish a radiculopathy from other lesions when the physical examination findings are unclear. Although EMG is very sensitive and specific, normal EMG results in a patient with signs and symptoms consistent with a radiculopathy do not exclude the diagnosis of radiculopathy. For example, a pure sensory lesion at the root level will have negative EMG and sensory nerve conduction study results. In this study, IRCE is also a kind of radiologic finding. Nevertheless, it has a pathological meaning of venous hypertension which induces axonal injuries. The validity of IRCE might be considered as somewhere between DCSA and EMG. Additionally, IRCE includes changes from all sensorimotor rootlets and has also the virtue of simplicity and good visualization compared with EMG in the selection of symptomatic level in lumbar central stenosis. These advantages will be able to benefit clinicians immediately for performing pain interventions and planning a minimally invasive surgery. Considering many aspects, IRCE may be complementary to neurologic examination, DCSA, and EMG results in patients with less obvious clinical complaints.

Reviewer 2 Report

Dear author,
your manuscript addresses a very interesting topic and presents new measurement methods for the quantification of spinal canal stenosis. The manuscript is clearly structured, although some readers may have trouble with the many abbreviations that disturb the reading flow. The scientific novelty and the potential clinical benefit is obvious.

I only have two main remarks:

1. Since neither graduation nor stratification, but simply cut-off values are proposed, I suggest changing the title to, for example, "Quantification of lumbar spinal canal stenosis by quantitative contrast-enhanced magnetic resonance imaging".

2. Venous congestion is initially plausibly hypothesized as the cause of the claudicatio spinalis symptoms and its exertion-dependent character. How does this fit together with the lying MRI examinations? Please discuss standing MRIs or post-exertional MRIs as a possible outlook.

Author Response

Point 1: Since neither graduation nor stratification, but simply cut-off values are proposed, I suggest changing the title to, for example, "Quantification of lumbar spinal canal stenosis by quantitative contrast-enhanced magnetic resonance imaging".

Response 1: Thank you for suggesting the appropriate title. I agree with your opinion. I changed my title to "Quantification of lumbar spinal canal stenosis by quantitative fat-suppressed contrast-enhanced magnetic resonance imaging".

Point 2: Venous congestion is initially plausibly hypothesized as the cause of the claudicatio spinalis symptoms and its exertion-dependent character. How does this fit together with the lying MRI examinations? Please discuss standing MRIs or post-exertional MRIs as a possible outlook. 

Response 2: Thank you for commenting about upright MRI or post-exertional MRI as a possible outlook. A paragraph given below was added in the discussion.

In conventional MR systems, the patient is scanned in the supine position. Venous congestion as the cause of neurogenic claudication cannot be fully evaluated. Dynamic compression during walking or repetitive exertional events cannot be examined. Nevertheless, it seems that venous enhancement patterns (PVCE, DVCE, IRCE) can be shown at rest on the fat-suppressed contrast-enhanced T1-weighted images. This may be because static compression by circumjacent bone and soft tissue is enough to develop venous hypertension. This means that venous hypertension and neuronal damage are in an advanced stage. From the physiological point of view, upright and post-exertional MRI will be the best method to evaluate the venous congestion in incipient stage (not visible on supine MRI but visible on upright MRI). Unfortunately, there are some compromises until now. While traditional supine scanners have a high strength magnetic field (generally 1.5T, but sometimes up to 3T), upright and open MRI scanners have a lower field strength (0.2-0.6T) with some compromise in image quality. Besides, fat-suppressed contrast-enhanced T1-weighted images may have a problem relating to field inhomogeneity due to this field strength. The constant improvement in both coil technology and software will have narrowed the gap in image quality between low and higher field systems.